# Selective Assembly of TRPC Channels in the Rat Retina during Photoreceptor Degeneration

**DOI:** 10.3390/ijms25137251

**Published:** 2024-06-30

**Authors:** Elena Caminos, Susana López-López, Juan R. Martinez-Galan

**Affiliations:** 1Department of Medical Science, Medical School of Albacete, Instituto de Biomedicina (IB-UCLM), University of Castilla-La Mancha, 02008 Albacete, Spain; susana.lopezlopez@uclm.es (S.L.-L.); juanramon.martinez@uclm.es (J.R.M.-G.); 2Consejo Superior de Investigaciones Científicas, and Research Unit, Complejo Hospitalario Universitario de Albacete, 02008 Albacete, Spain

**Keywords:** calcium channels, neurodegeneration, retinitis pigmentosa, STIM1, transient receptor potential canonical, TRPC1/5

## Abstract

Transient receptor potential canonical (TRPC) channels are calcium channels with diverse expression profiles and physiological implications in the retina. Neurons and glial cells of rat retinas with photoreceptor degeneration caused by retinitis pigmentosa (RP) exhibit basal calcium levels that are above those detected in healthy retinas. Inner retinal cells are the last to degenerate and are responsible for maintaining the activity of the visual cortex, even after complete loss of photoreceptors. We considered the possibility that TRPC1 and TRPC5 channels might be associated with both the high calcium levels and the delay in inner retinal degeneration. TRPC1 is known to mediate protective effects in neurodegenerative processes while TRPC5 promotes cell death. In order to comprehend the implications of these channels in RP, the co-localization and subsequent physical interaction between TRPC1 and TRPC5 in healthy retina (Sprague-Dawley rats) and degenerating (P23H-1, a model of RP) retina were detected by immunofluorescence and proximity ligation assays. There was an overlapping signal in the innermost retina of all animals where TRPC1 and TRPC5 physically interacted. This interaction increased significantly as photoreceptor loss progressed. Both channels function as TRPC1/5 heteromers in the healthy and damaged retina, with a marked function of TRPC1 in response to retinal degenerative mechanisms. Furthermore, our findings support that TRPC5 channels also function in partnership with STIM1 in Müller and retinal ganglion cells. These results suggest that an increase in TRPC1/5 heteromers may contribute to the slowing of the degeneration of the inner retina during the outer retinal degeneration.

## 1. Introduction

Retinitis pigmentosa (RP) is a group of genetic degenerative diseases that cause progressive loss of photoreceptors, leading to complete loss of vision. The P23H line 1 (P23H-1) transgenic rats carry a common rhodopsin mutation (proline-to-histidine substitution), generating an experimental model of neurodegeneration with features of RP [1,2]. In these animals, inner retinal morphology and function are apparently preserved as photoreceptor degeneration progresses and retinal ganglion cell (RGC) loss is not significant up to one year of postnatal life [1,3]. In contrast, the astrocytes and Müller cells, which provide support for retinal neurons and maintain normal retinal function, exhibit reactive gliosis and marked hypertrophy with upregulation of intermediate filament proteins and metabolic and gene expression alterations directly related to photoreceptor death [1,3,4,5,6,7,8]. This progressive retinal disruption largely matches the clinical findings analyzed in humans with RP [9].

Neuronal survival in the innermost retina has put the focus of neuronal rescue and repair therapies on the bipolar cells, ganglion cells, and Müller cells by optogenetics techniques, among others [10,11,12,13,14]. In this region, baseline intracellular calcium levels in neuronal and glial cells are significantly higher in P23H rats than in healthy retinas [15]. Indeed, in several rat models of autosomal dominant RP, photoreceptor death was associated with increased intracellular calcium [16].

Several members of the transient receptor potential canonical (TRPC) channel family display different distributions and functions in health during retinal pathological processes (see reviews in [17,18]). TRPC channels are involved in the calcium ion entry triggered by metabotropic receptor stimulation, depletion of intracellular calcium stores, and/or interactions with other proteins. Seven TRPC channels have been identified (TRPC1, TRPC2, TRPC4/5, and TRPC3/6/7) and divided into four subgroups with similar amino acid sequences and mechanisms of action within each group. They all function as calcium-permeable, non-selective cation channels that form homo- or hetero-meric complexes [19,20]. TRPC1 and TRPC5 are good candidates to form heteromers in the rat retina, as we show in this study.

The function of TRPC1 is linked to store-operated channels (SOCs) in addition to other activation mechanisms associated with (1) the activity of mGluRs in the plasma membrane, (2) IP3Rs in the reticulum endoplasmic membrane, (3) the sensor stromal interacting molecule 1 (STIM1) in the reticulum membrane activated after calcium store depletion, and (4) the polymerization of the channel proteins Orai1 [21,22,23]. In the rat retina, the TRPC1 channel may be activated by different mechanisms depending on cell type and probably on the cellular context [24]. RGCs and Müller cells possess all the necessary machinery to respond to retinal stressful situations through the TRPC1 channels previous activation of metabotropic receptors, inositol trisphosphate receptors, STIM1, and even ryanodine receptors [17,25,26,27,28,29,30,31,32]. TRPC5 channels are primarily activated by receptors coupled to Gq and the phospholipase C pathway and they are also activated upon depletion of intracellular calcium stores, which requires other partner proteins, such as TRPC1, STIM1, and Orai1, and other stimuli, such as mechanical stimulation (membrane stretch) and temperatures between 25 and 37 °C. Moreover, TRPC5 functions in partnership with numerous proteins, including TRPC1 and TRPC4 [33,34,35,36].

TRPC1 has been identified as a source of cytosolic calcium in dissociated mouse Müller cells, with a relevant function in response to stress mechanisms like glaucoma [26,29]. Moreover, this channel is widely distributed in the rat retina and its expression increases in the cells of the inner retina during photoreceptor degeneration by RP [24]. In experimental models of Parkinson’s disease, the dopaminergic neurons of TRPC1^-/-^ mice have increased rhythmic activity, leading to neurotoxicity and death. Meanwhile, the overexpression of TRPC1 in dopaminergic neurons protects them from death [37,38,39]. On the other hand, TRPC5 is also highly expressed in neuronal tissue and has been implicated in the regulation of neurite length and growth [40,41]. Axons of RGCs and neurites of hippocampal neurons were longer when TRPC5 was somehow inactivated [40,41]. Moreover, the TRPC5 channel contributes to the death of RGC when abnormal intraocular pressure changes occur, such as glaucoma [35]. Here, we analyze the distribution of TRPC5 during photoreceptor degeneration in the P23H-1 rat model to investigate its relationship with TRPC1.

For this purpose, we have considered the possibility that the TRPC1 and TRPC5 channels are involved in the maintenance of retinal calcium homeostasis, forming the heterodimer complex with special relevance during retinal degeneration due to RP. TRPC5 has even been shown to physically interact with TRPC1 in brain neurons and in PC12 cells [42,43,44]. When TRPC1 forms heteromeric channels, it acts as a negative regulator for TRPC4 and TRPC5 channels [45]. One of the more relevant consequences is that the calcium permeability is much lower when TRPC1 and TRPC5 form a heteromeric complex than when TRPC5 is a homomeric channel [45]. Both channels are distributed throughout the inner retina. While TRPC1 is expressed in astrocytes, Müller cells, and RGCs, among others, of the rat retina and in cultured mouse retinal Müller cells [24,25,26,46], TRPC5 has also been localized in RGCs and in Müller cell cultures of the mouse retina [29,35], but it has not been described in the rat retina until the present work.

The present study also sought to determine whether TRPC1 and TRPC5 function as heteromeric complexes (TRPC1/5) in the normal rat retina and during photoreceptor degeneration due to RP. Furthermore, the aim was to elucidate the role of calcium in the inner retina as the outer retina degenerates. We found that a physical interaction exists between TRPC1 and TRPC5 and that this interaction is promoted by retinal degenerating mechanisms. TRPC1 activity may be relevant in RGC and glial cells when TRPC1/5 heteromeric complexes are formed in the retina of the P23H rat. It is important to know the survival conditions of the cells of the inner retina because their functioning is essential to maintain the activity of the visual cortex, even if the person is already totally blind.

## 2. Results

### 2.1. Identity of TRPC1 and TRPC5 Channel Assembly in Retinal Cells

To determine whether TRPC1 and TRPC5 channels can form homo- or hetero-meric complexes in rat retinas, firstly, we analyzed the possible co-existence of both channels by double immunohistochemistry and, subsequently, we analyzed the possible physical interaction between TRPC1 and TRPC5 (TRPC1–TRPC5) by proximity ligation assay (PLA), in healthy rat retinas and retinas with photoreceptor degeneration. While TRPC1 is widely distributed in all retinal layers (Figure 1A–E), TRPC5 immunoreactivity is detected in the inner retinal of SD and P23H rats (Figure 1F–J). TRPC1 immunoreactivity patterns have been reported previously [24] with staining in specific cell somata of the retinal pigment epithelium, inner nuclear layer (INL), ganglion cell layer (GCL), and nerve fiber layer (NFL), as well as in processes in the plexiform layers of the rat retinas (first column of images in Figure 1). RGCs and glial cells were immunoreactive for TRPC1 antibody. TRPC5 immunoreactivity had a more restrictive distribution than TRPC1 (second column of images in Figure 1). It was detected in the inner plexiform layer (IPL), GCL, and NFL in all the retinas used. RGCs and glial cells immunoreactive for TRPC5 could be identified in sections and whole retinas (boxes in Figure 1F–I) of all animals. We did not detect specific variations in TRPC5 immunoreactivity between both strains SD and P23H at different ages beyond those related to the degenerative process. The overlapping signal between TRPC1 and TRPC5 was observed in the GCL and NFL (Figure 1K–O). These labeling and co-localization distribution patterns were localized in the RGCs and glial cells and remained in P23H rat retinas even after photoreceptor degeneration (detail in Figure 1E,J,O).

The TRPC1–TRPC5 PLA signal reproduced the co-localization immunolabeling patterns in the inner retina, as described above (Figure 2). Red fluorescent dots distributed in the GCL and NFL of all retinas tested at each age were quantified. No red PLA signaling was detected in any of the controls (Figure 2E). No apparent differences were observed between SD and P23H rat strains at P30 (Figure 2A,C). The number of red dots became progressively larger with age and increased in both rat strains, SD (Figure 2A,B) and P23H (Figure 2C,D). Finally, as photoreceptor degeneration progressed, red PLA signaling increased, as can be observed in the P23H retinas at P240 (Figure 2B,D). Maxima interaction between subunits of both channels was found in P23H retinas at P240. These observations were confirmed by quantification analysis in the inner retinal layers (Figure 2F) through the Kruskal–Wallis test that resulted in a significance with *p* < 0.001 (H = 44.12). Moreover, Dunn’s test performed to evaluate the differences between the pairs of mean values indicated that the TRPC1–TRPC5 interaction significantly increased with age in the inner retina of SD rats (*p* < 0.01) and more acutely with the loss of photoreceptors in P23H rat retinas (*p* < 0.0001).

To better understand the mechanism of action of TRPC5 and TRPC1 channels in the retina, relative TRPC5 expression levels were analyzed by Western Blot (Figure 3) in normal and RP retinas at P30 and P240. The assays showed the presence of TRPC5 with a single band of around 111 kDa in the extracts from SD and P23H rat retinas (Figure 3A). The Kruskal–Wallis test (H = 2.08; *p* = 0.55) and the subsequent Dunn’s test to evaluate differences between groups showed no statistical differences in protein levels between the SD and P23H strains (Figure 3B). Furthermore, there were no statistical differences in protein levels between animals of the same strain and different ages. These results align with those observed in the immunocytochemistry study, where no differences in TRPC5 distribution were observed between all groups of animals studied, despite the glial reaction present in the P23H retinas, which is analyzed below.

### 2.2. TRPC1, TRPC5, and STIM1 in the Innermost Layers of the Rat Retina

Considering that TRPC1 and TRPC5 channels may gate in a store-dependent manner, which requires other partner proteins, such as STIM, we also seek here specific co-localization between TRPC1 and STIM1 and between TRPC5 and STIM1. First, we identify a specific localization of STIM1 in the inner retina using an anti-GFAP antibody (Figure 4), which is expressed by astrocytes of the normal retina and also by Müller cells in response to degeneration or retinal injury [47]. In SD and P23H retinas, STIM1 immunoreactivity was distributed throughout the inner retina, occupying astrocytes, Müller cells, and RGCs, with a greater distribution in the Müller cell processes as the retina degenerated (Figure 4A–D’). The STIM1/GFAP co-localization was increasing in degenerating retinas probably because of the gliotic reaction (Figure 4C–D’). Similar results were found analyzing the co-localization between TRPC5 and GFAP (Figure 4E–H’), with co-localization in astrocytes of healthy retinas and with high immunoreaction in Müller cells of P23H retinas at any age analyzed. Moreover, TRPC5 immunoreactive cells in the GCL, which were not immunopositive for GFAP (Figure 4E–F’), correspond to RGCs where TRPC5 also co-localized with TRPC1 (Figure 1). In whole retinas (Figure 4I–K), it was easier to appreciate glial cells immunoreactive for TRPC1, TRPC5, and STIM1 in older P23H retinas. The TRPC1/GFAP co-localization immunoreactivity was previously reported in glial cells of rat retinas [24] (Figure 4I and Appendix A).

Finally, we looked for possible co-localization of STIM1 with TRPC1 or TRPC5 (Figure 5 and Figure 6). We only occasionally found STIM1 in cells immunolabeled for TRPC1 in the GCL (Figure 5A). These cells had the morphology of RGCs. This distribution pattern is repeated even when the retinas no longer have photoreceptors (Figure 5B). The results of the wholemount retinas analysis indicated that TRPC1 and STIM1 immunoreactivities coexisted, but rarely co-localized, in the cytoplasm of RGCs (Figure 5C). Furthermore, STIM1 was also distributed in glial processes around RGC somata that were immunoreactive to TRPC1. In contrast, there are numerous cells and elements immunoreactive for TRPC5 and STIM1 in the GCL and NFL in all retinas (Figure 6). TRPC5 and STIM1 immunoreactivity were clearly co-localized in RGCs and astrocytes of SD retinas (Figure 6A,C) and in RGCs, astrocytes, and Müller cells of P23H retinas (Figure 6B).

## 3. Discussion

### 3.1. TRPC1–TRPC5 Interaction in Response to Retinal Degeneration

TRPC1 and TRPC5 are ion channels tightly linked to the nervous system activity. There is good evidence that they differ in their distribution and physiological effects on the retina. We show that both channels physically interact and may, therefore, function as TRPC1/5 heterodimers in the rat retina. While TRPC1 expression and the TRPC1–TRPC5 interaction increased during retinal degeneration due to RP, the relative levels of TRPC5 did not change with age or due to photoreceptor degeneration. According to our data, TRPC1 markedly responds to retinal degenerative processes. Furthermore, it is plausible that at least part of the activation of TRPC channels in inner retinal cells is related to STIM1 proteins, mainly in RGCs and glial cells [17,25,26,29], as well as other activation mechanisms depending on the retinal cell type [24].

In the retina, TRPC1 and TRPC5 differ significantly in their distribution and physiology, but both have related effects. TRPC1 expression is ubiquitous in the retina, whereas TRPC5 expression is concentrated in the inner retina, mainly in Müller cells, astrocytes, and ganglion cells of the mouse and rat retina [17,24,25,26,29,35]. There are several pathological models in which the outcome of the heteromeric activity of TRPC1 and TRPC5 has been studied. TRPC1 serves as a suppressor of the pressure-induced gliosis response, suggesting that it may play a neuroprotective role in such pathological circumstances. In a mouse model of Parkinson’s disease, TRPC1 overexpression prevents the development of the disease and these protective effects disappear in the TRPC1 knockout [37,38,39]. In contrast, TRPC5 is an important negative regulator of RGC axonal growth and neurite remodeling, which could lead to cell death when abnormal intraocular pressure occurs [29,35]. The excess calcium entering through TRPC5 then intervenes in the remodeling of axons and dendrites, promoting cell death. Similar mechanisms have been described in mouse hippocampal neurons, where inactivation of TRPC5 results in longer neurites and growth cones with longer filopodia [40]. In Huntington’s disease cell lines, increased calcium influx through TRPC5 causes cell death, whereas, in wild-type cell lines, calcium influx through TRPC5 was reduced when both TRPC1 and TRPC5 formed a heteromeric complex [45,48]. Therefore, the various pathological models demonstrated the opposite effects of TRPC1 and TRPC5 in neurodegenerative diseases and neurite outgrowth [43,49]. It has been proposed that the overexpression of TRPC1 and inhibition of TRPC5 may help reduce the neurodegeneration effects.

While the loss of TRPC1 channels does not impact the kinetics of cones and rods [46], studies have shown that in animals with RP, there is an increase in immunoreactivity and activation of TRPC1 in photoreceptors when homeostasis conditions are altered by the rod outer segment loss [24,50]. In P23H animals, the mutation causes rhodopsin misfolding and the induction of the unfolded protein (UPR) response, which leads to persistent endoplasmic reticulum stress on the inner segment of photoreceptors [16]. This has been considered to be an adaptive response to try to overcome reticulum stress, attempting to modulate calcium homeostasis by mitochondrial reserves and in the endoplasmic reticulum [51,52,53]. The evidence for this mechanism is based on the observation that, despite the fact that the fate of these cells is death, the inner segment takes longer to degenerate and the threshold of the b-wave remains constant with the considerable thinning of the outer nuclear layer [4].

When this process occurs in the outer retina, there is an increase in TRPC1 expression and its interaction with TRPC5 in the inner retina of P23H rats. Our experiments cannot prove that the increase in TRPC1/5 heteromeric forms during photoreceptor degeneration has an effect on inner retinal maintenance, but we can show that events occur that are consistent with the results of many other studies. Indeed, the TRPC1/5 heteromers have been shown to have increased activity when the intracellular calcium concentration is higher than basal levels [44,45,54]. These are the circumstances that occur in the retina of P23H rats, where basal calcium levels in the cells of the inner retina are higher than in the retinas of healthy SD rats [15]. One possible explanation is that TRPC5 homomers conduct relatively large Na^+^ and Ca^2+^ inward currents, whereas the inward current through heteromeric TRPC1/5 channels is weakly inwardly rectifying [44,45]. Thus, in the RP model used in our study, TRPC1 could also slow down the TRPC5 activation, thereby helping to preserve inner retinal cells. We propose that this would result in calcium levels in the inner retinal that are not as toxic as when TRPC5 acts as a homomer. Then, retinal degeneration in P23H rats does not compromise the activity of the visual cortex for an extended period of time. This is evidenced by the fact that cortical potentials evoked by electrical stimulation remain stable and presynaptic (thalamocortical terminals) and postsynaptic elements (dendritic spines in layer V) undergo changes only when retinal degeneration is at an advanced stage [55,56]. Although glutamate release is impaired, the maintenance of calcium concentration keeps the cells of the inner retina active. A model of the effects of the TRPC1/5 heteromer in P23H retinas is proposed in Appendix A. All previous studies, together with our results, open the possibility of a further series of experiments needed to corroborate whether the TRPC1–TRPC5 interaction could be used as a mechanism to decelerate inner retinal degeneration, thus providing more treatment options for RP patients.

### 3.2. TRPC1, TRPC5, and TRPC1/5 Channels Focused in RGCs and Müller Cells

The rat retinal cells in which the expression of TRPC1, TRPC5, and STIM1 was detected were astrocytes, Müller cells, and RGCs and the overlap between these proteins was more easily detected in retinas with active gliosis due to a process of neuronal degeneration. Müller cells are the first cells in the retina to respond to stressors and environmental, genetic, and signaling factors. They have complex mechanisms to maintain gliosis and high calcium levels, as well as metabolic functions to support neural repair and cell survival in the retina [57]. In Müller cell endfeet, there is a strong expression of TRPC1 proximal to endoplasmic reticulum cisternae with inositol triphosphate receptors, ryanodine receptors, STIM1, STIM2, and Orai [17,25,26,29,46]. However, the integrity of the inner retina in RP also depends on the preservation of RGCs, which are the most resistant to degeneration and are responsible for gathering important information, such as maintaining circadian cycles. In this work, we show the interaction of TRPC1 and TRPC5 with the possibility that both channels form heteromers in the retina. TRPC1/5 heteromeric channels are known to be coupled to G-protein-coupled receptors [45] and group I metabotropic glutamate receptors and STIM1 are present in rat RGCs [32,58,59]. Therefore, Müller cells and RGCs would have the necessary machinery to slow down TRPC5 activation and then delay degeneration through the intervention of TRPC1 to form TRPC1/5 heteromeric channels in retinas with RP.

In summary, this study demonstrates a physical interaction of TRPC1 and TRPC5 in the rat retina and this interaction is greater when the retina has a large loss of photoreceptors. To arrive at these results, we also describe for the first time the immunodistribution of TRPC5 in the RGCs and glial cells of the rat retina. The TRPC1–TRPC5 interaction was concentrated in elements of the inner retina where RGCs and Müller cells are most resistant to degeneration due to RP and they have the machinery necessary for the activity of these channels. Many other studies in different neuropathological models have shown that TRPC1 is a negative regulator of TRPC5 channels and that neuroprotection is promoted by the presence of the TRPC1/5 heteromers. In the retina, TRPC1 and TRPC1/5, but not TRPC5, function in response to the degenerative mechanisms associated with RP. It is plausible to think that the predominance of TRPC1/5 heteromers over homomeric forms of TRPC5 slows down inner retinal degeneration in P23H rats. It would be useful to know the evolution of RP following the blockade of TRPC1 or TRPC5 in experimental models of RP to determine the potential therapeutic efficacy of this approach. This objective requires further investigation.

## 4. Materials and Methods

### 4.1. Experimental Animals and Ethical Approval

P23H-1 homozygous albino rats were used in this study as an experimental model of neurodegeneration. These animals underwent gradual, fast photoreceptor loss, which is characteristic of autosomal dominant retinitis pigmentosa [1]. These rats were first provided by Dr. Matthew LaVail (Beckman Vision Center, San Francisco, CA, USA) and then they were bred in a colony at the Animal House of the University of Castilla-La Mancha (Medical School, Albacete, Spain). The background of the P23H rat is the strain Sprague-Dawley (SD, Charles River Laboratories, Barcelona, Spain). SD rats were used for wild-type healthy controls. A total of 48 female rats, aged 30 and 240 postnatal days (P), of both stains (SD and P23H) were used and distributed among the different tests (28 animals for immunocytochemistry and PLA, and 20 animals for WB). The use of only female rats was employed in this study due to the existence of differences in eye size that can potentially lead to microscopic variations in cell density. Consequently, variations in immunostaining and signal density may be observed in the interaction experiments.

All protocols were approved by the UCLM Ethics Committee for Experimental Animal Welfare, in accordance with European and Spanish legislation (Directive 2010/63/UE, amended by Regulation (EU) 2019/1010, and RD 53/2013).

### 4.2. Retinal Processing

Rat retinas from seven animals per age (P30 and P240) and strain (SD and P23H-1) were anesthetized and transcardially perfused according to previously published protocols [24]. Cryosections from 16 retinas were used for immunocytochemistry and proximity ligation assays. Additionally, 40 retinas were isolated and processed in toto for immunocytochemistry. All animals were anesthetized with a mixture of ketamine (100 mg/kg, Parke-Davis, Alcobendas, Spain) and 2% xylazine (10 mg/kg, Dibapa, Barcelona, Spain) and transcardially perfused with 0.9% saline solution and 4% paraformaldehyde in 0.1 M phosphate buffer (PB). Eyes were dissected and postfixed for 4 h after removing the lens. To obtain retinal sections, eyes were previously cryoprotected in 30% sucrose and frozen with liquid nitrogen. The frozen blocks were stored at −80 °C until use. In addition, whole retinas were isolated after postfixing and stored in a glycerol solution at −20 °C until use.

### 4.3. Double Immunocytochemistry

Retinal sections of 18 mm thickness from four animals per strain and age were cast on slides and washed in PB saline, pH 7.3 (PBS), containing 0.25% Triton X-100 (PBST). Slides were then incubated for 1 h at room temperature (RT) with the blocking solution containing PBST and 5% bovine serum albumin (BSA, Fraction V, Sigma-Aldrich, Steinheim, Germany). To identify specific retinal cells and to look for co-localization between proteins, sections were incubated in a mixture containing two primary antibodies (Table 1) in PBST-BSA for 14–17 h. The immunoreactivity of the primary antibodies was visualized using anti-mouse and anti-rabbit, IgG (H + L), and secondary antibodies coupled to Cy2, Cy3, or Cy5 (1:200, Jackson ImmunoResearch, Baltimore Pike, PA, USA) for 1 h. After the incubations with secondary antibodies, retinal sections were washed in PBST, air-dried in the dark, and covered with Fluoroshield mounting medium (Sigma-Aldrich) containing 4′,6-diamidine-2′-phenylindole dihydrochloride (DAPI). In the whole retinas, primary antibodies were performed for 2 days and secondary antibodies for 3 h. The experiments were conducted in triplicate for each combination of primary antibodies.

Images were acquired by a laser confocal microscope (Zeiss LSM 710, Jena, Germany) and analyzed by the ZEN Blue Edition software (Carl Zeiss Microscopy, Jena, Germany). Artworks were organized using Adobe Photoshop© (Ps 1990–2023 Adobe V25.1.0). The controls for the immunocytochemistry included: (1) The incubations with the TRPC1 antibody pre-absorbed with the TRPC1 blocking peptide; (2) incubations with only secondary antibodies, and (3) incubations with secondary antibodies that did not recognize the host species in which the corresponding primary antibodies were obtained. None of the controls gave specific labeling.

### 4.4. Proximity Ligation Assays (PLAs)

To determine whether TRPC1 forms heteromeric channels with TRPC5 in the retina, protein interactions were detected by in situ proximity ligation assays (PLAs) using Duolink in situ PLA kits (Merck, Darmstadt, Germany with modifications to the protocol used by Caminos and collaborates [15]. Retinal slides from four animals per strain and age were used. Double immunocytochemistry was performed with primary antibodies polyclonal anti-TRPC1 [1:400] and monoclonal anti-TRPC5 [1:400], with incubation for 14–17 h at RT, in a humid chamber. PLA probe anti-rabbit PLUS and PLA probe anti-mouse MINUS were used as secondary antibodies, diluted 1:5 in Duolink Antibody Diluent and incubated for 60 min at 37 °C. Retinal sections were then washed with Wash Buffer A (PLA kit, Merck) and incubated in the hybridization solution containing 1:40 Ligase for 60 min at 37 °C. After incubation, slides were washed with Wash Buffer A. The amplification solution containing the fluorescent oligonucleotides and the polymerase diluted at 1:90 was applied to the sections. Finally, the slides were washed with Wash Buffer B (PLA kit, Merck) and mounted with a medium containing DAPI (Sigma-Aldrich). Signal interaction was analyzed by identifying red fluorescent dots under a confocal microscope. Technical controls consisted of a cross-reaction between secondary antibodies and reactions without primary antibodies. PLA signals were absent in all control sections.

A minimum of 28 images per experimental condition were analyzed from two independent replications of each experiment. They were acquired using a Zeiss laser scanning microscope and analyzed by the ZEN Blue Edition software (Carl Zeiss Microscopy, Jena, Germany). To identify fluorescent signals, the 580/604 nm laser line was used for excitation/emission wavelength, respectively. To identify retinal and cell layers, a DAPI 405/448 nm excitation/emission wavelength laser line was used. All the settings were kept constant in all the experiment groups for image acquisition purposes. One image from each stack was automatically selected by the ZEN software and PLA signals (red fluorescent dots) were quantified using the free image software ImageJ (Fiji; https://imagej.softonic.com/; Accessed online: 30 June 2024). Statistical analysis was performed by a Kruskal–Wallis test followed by Dunn’s multiple comparison test using GraphPad Prism 5.0 (GraphPad Software, La Jolla, CA, USA). Values are expressed as arithmetic mean ± standard error of the mean (SEM).

### 4.5. Western Blots

Retinas from at least 5 animals per age (P30 and P240) of each rat strain (SD and P23H) were employed. Retinas were isolated and homogenized in 800 mL of ice-cold commercial RIPA lysis buffer (Sigma-Aldrich) supplemented with a cocktail of protease inhibitors and II phosphatase inhibitor (Sigma-Aldrich) for 30 min at 4 °C and centrifuged at 12,000× *g* for 15 min at 4 °C. After determining protein concentration, 40 µg of total protein extracts were separated in 10% polyacrylamide gels for over 100 min at 30 mA. The gels were transferred to PVDF membranes (Hybond-C Extra, Amersham Biosciences) using a wet blotter (Bio-Rad, Hercules, CA, USA) for 120 min at 350 mA. The membranes were blocked and were incubated with anti-TRPC5 (see Table 1) in TBST-BSA overnight at 4 °C. A secondary antibody anti-mouse IgG HRP-conjugated 1:2000 (Jackson ImmunoResearch) was used for the detection of TRPC5. Immunoblotting signals were detected with the Luminescent Image Analyzer LAS-mini 4000 system (Fujifilm, Tokyo, Japan). Finally, an anti-GAPDH antibody (Table 1) was used as the loading control. The reaction control included the incubation of membranes without primary antibodies.

Densitometric analysis was performed using ImageJ (Fiji software, V 1.8.0) and proteins were normalized to the loading control GAPDH. Statistical data were determined from six different samples per age and strain using GraphPad Prism 5.0. Statistical significance between samples was analyzed by the Kruskal–Wallis test followed by Dunn’s multiple comparison test.

## Figures and Tables

**Figure 1 ijms-25-07251-f001:**
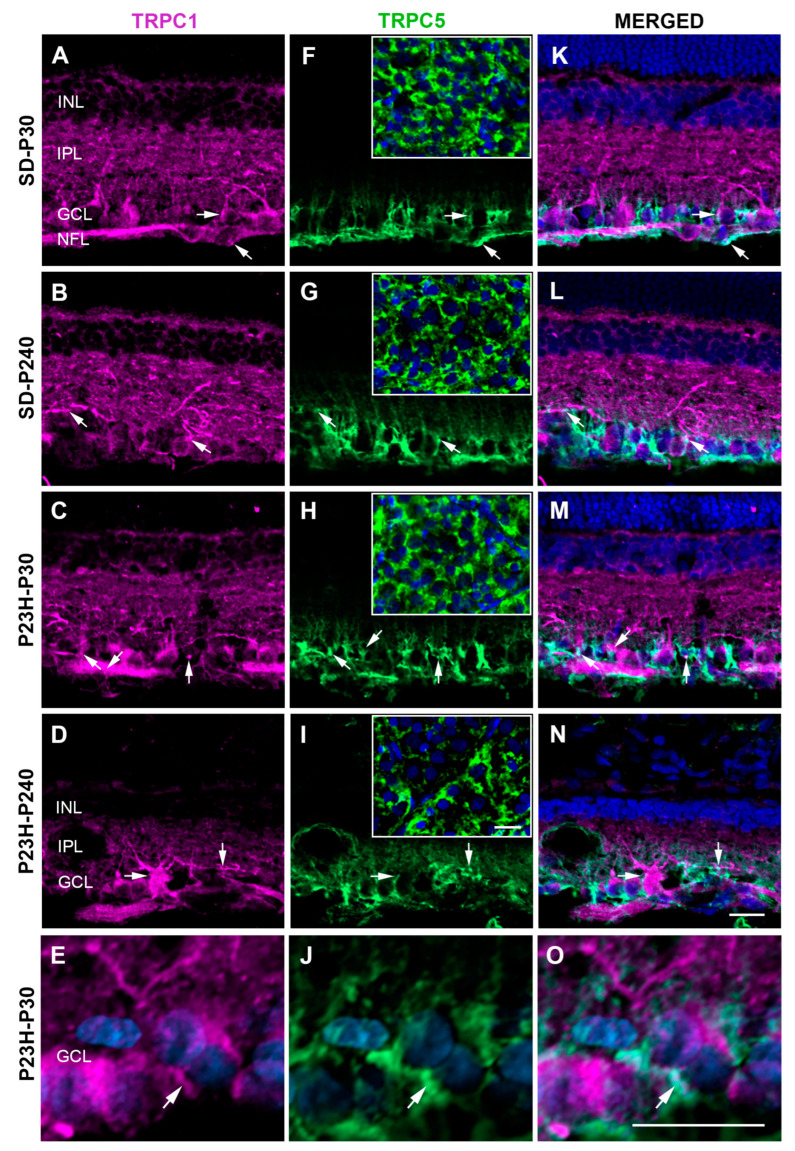
This is a figure. Schemes follow the same formatting. Confocal images showing TRPC1 and TRPC5 immunoreactivities in SD and P23H rat retinas. (**A**–**E**) Retinal sections with TRPC1 immunoreactivity (magenta) prominently distributed throughout all retinas in all animals. (**F**–**J**) Confocal images showing TRPC5 immunoreactivity (green) distributed in the inner retina of all animals. The boxes in the second column contain images of whole retinas taken at the level of the ganglion cell layer. (**K**–**O**) Merged images showing TRPC1/TRPC5 co-localization (arrows) distributed throughout the inner retina. (**E**,**J**,**O**) Magnification of the GCL showing a detail of TRPC1 and TRPC5 overlapping in the P23H retina at P30. Nuclei are stained with DAPI (blue). GCL, ganglion cell layer; INL, inner nuclear layer; IPL, inner plexiform layer; NFL, nerve fiber layer. Scale bars: 20 μm.

**Figure 2 ijms-25-07251-f002:**
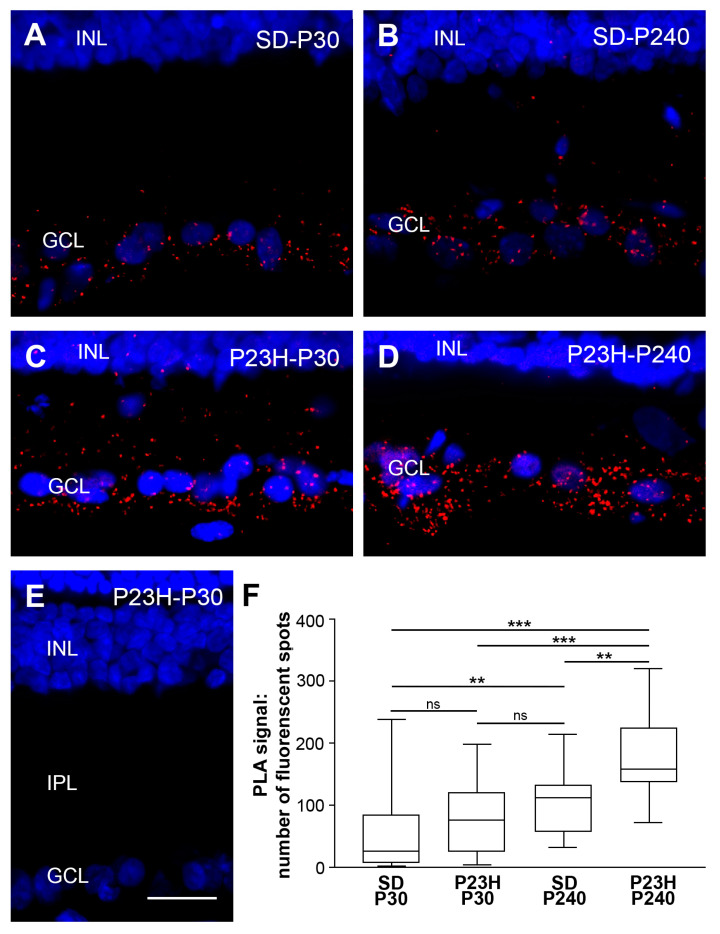
Representative images from retinal sections demonstrating the TRPC1–TRPC5 interaction (red dots) in the inner rat retina by proximity ligation assay. Nuclei are stained with DAPI (blue). The TRPC1–TRPC5 PLA signal was detected on the inner retinal layers of healthy retinas (**A**,**B**) and retinas with retinitis pigmentosa (**C**,**D**). (**E**) Representative image from control reaction where a cross-reaction between secondary antibodies did not show red PLA signals. (**F**) Quantification of the PLA signal, where the Y-axis is the number of red spots per 210 μm of retina. Statistically significant differences with *p* < 0.01 (**) and *p* < 0.001 (***), and no statistically significant differences (ns) according to Dunn’s test. GCL, ganglion cell layer; INL, inner nuclear layer; IPL, inner plexiform layer. Scale bar: 20 μm.

**Figure 3 ijms-25-07251-f003:**
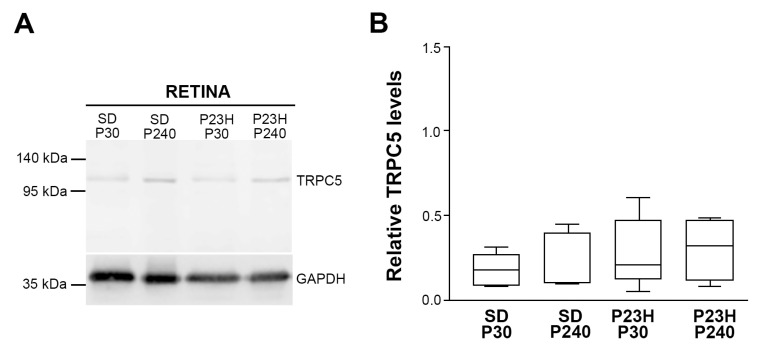
Relative TRPC5 protein levels in the control rat retinas (SD) and the retinas with photoreceptor degeneration (P23H-1) at different ages. GAPDH is the internal standard. (**A**) Western Blot showing TRPC5 in rat retinas. (**B**) Boxplot chart showing a comparison of the relative TRPC5 expression levels between the healthy retinas and P23H rat retinas. No statistically significant differences were observed between the groups, as determined by Kruskal–Wallis’s test and the subsequent Dunn’s test. Values are expressed as the ratio of the optimal density on the TRPC5 bands according to that on the GAPDH bands (TRPC5/GAPDH). Data were obtained from six independent determinations.

**Figure 4 ijms-25-07251-f004:**
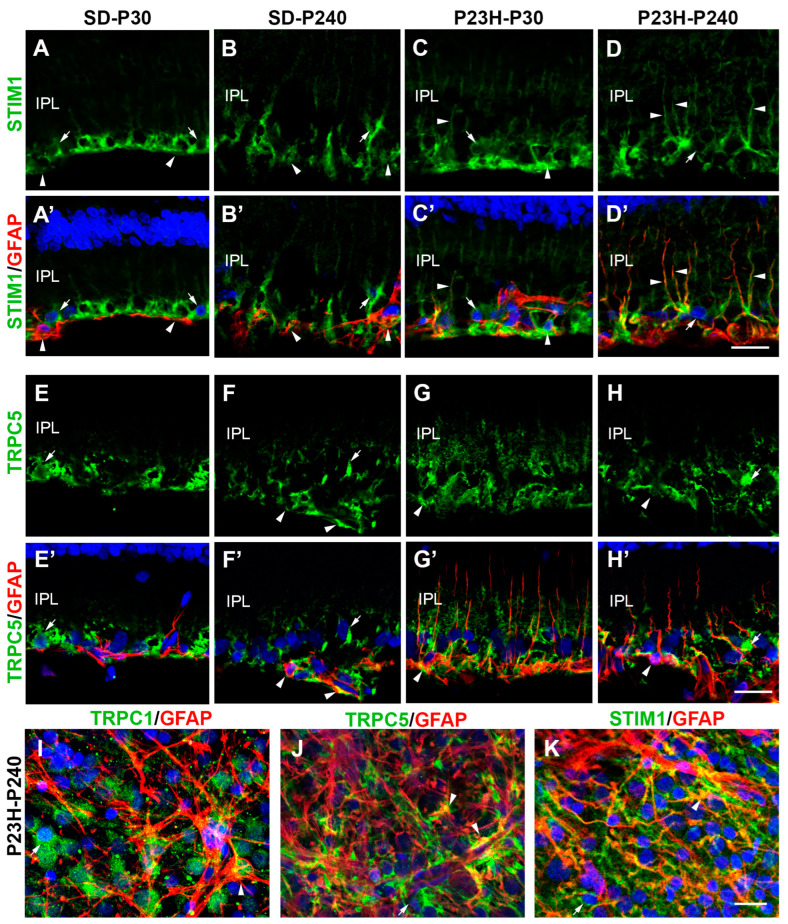
Identification of astrocytes and Müller cells immunoreactive for TRPC5 and STIM1 in SD and P23H rat retinas. (**A**–**D’**) Retinal sections immunostained for STIM1 (green) and GFAP (red). (**E**–**H’**) Retinal sections immunostained for TRPC5 (green) and GFAP (red). Overlap is observed in astrocytes and Müller endfeet and vertical processes (arrowheads). (**I**–**K**) Wholemounted retinas from aged P23H rats showing co-localization of GFAP (red) with TRPC1 (**I**), TRPC5 (**J**), and STIM1 (**K**). Images were taken at the level of the GCL and NFL, where overlap is observed in Müller cells and astrocytes (yellow). Retinal ganglion cells are immunoreactive for TRPC1, TRPC5, and STIM1 (arrows). Nuclei are labeled with DAPI (blue). IPL, inner plexiform layer. Scale bar: 20 μm.

**Figure 5 ijms-25-07251-f005:**
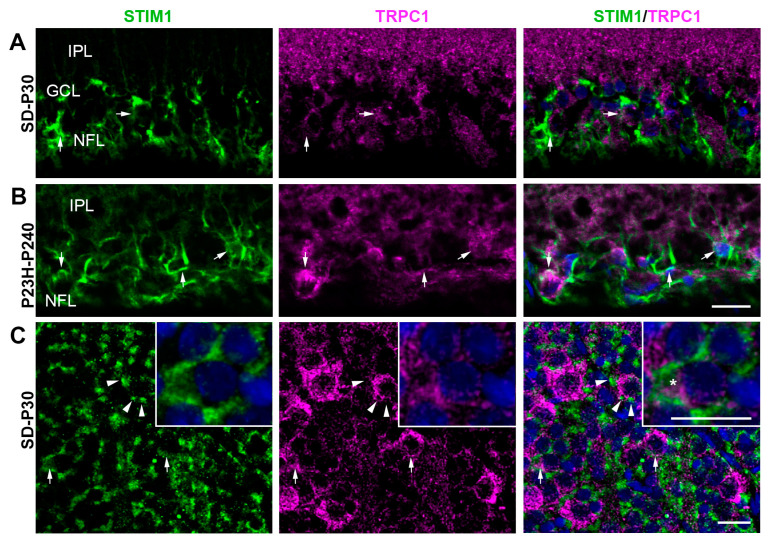
Confocal images showing STIM1 and TRPC1 immunolocalization in retinal sections of the inner retina from SD (**A**) and P23H (**B**) rat retinas and in a whole retina from a healthy rat (**C**). Exceptional co-localization of STIM1/TRPC1 in retinal ganglion cells (arrows and magnification in boxes in (**C**), asterisk) and STIM1 immunoreactivity around RGCs (arrowheads). Nuclei are stained with DAPI (blue). GCL, ganglion cell layer; IPL, inner plexiform layer; NFL, nerve fiber layer. Scale bar: 20 μm.

**Figure 6 ijms-25-07251-f006:**
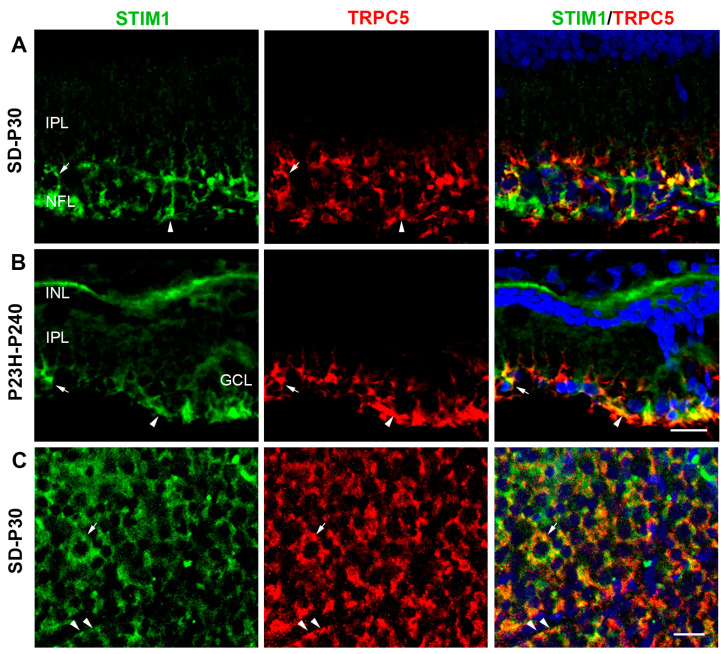
Confocal images showing TRPC5 and STIM1 immunolocalization in retinal sections of the inner retina from SD (**A**) and P23H (**B**) rat retinas and in a whole retina from a healthy rat (**C**). STIM1 (green) and TRPC5 (red) and the co-localization of STIM1/TRPC5 (yellow) are shown in the merged image. Extensive co-localization is observed in retinal ganglion cells (arrows) and glial cells (arrowheads). Nuclei are stained with DAPI (blue). GCL, ganglion cell layer; IPL, inner plexiform layer; NFL, nerve fiber layer. Scale bar: 20 μm.

**Table 1 ijms-25-07251-t001:** Features of the primary antibodies.

Antigen	Immunogen	Host/Mono-Polyclonal	Manufacturer	Dilution
GAPDH	Glyceraldehyde-3-phosphate dehydrogenase (whole molecule)	Mouse Monoclonal	ThermoFisher Scientific (Foster City, CA, USA), AM4300, Clon 6C5	WB ^1^: 1:8000
GFAP	Glial Fibrillary Acidic Protein isolated from cow spinal cord	Rabbit Polyclonal	Dako (Santa Clara, CA, USA), #Z0334	IC ^2^: 1:1000
GFAP	Glial Fibrillary Acidic Protein isolated from cow spinal cord	Mouse Monoclonal	Sigma (Steinheim, Germany), #G3893, Clone G-A-5	IC: 1:1000
STIM1	N-terminal of human STIM1	Rabbit Polyclonal	Proteintech (Manchester, UK), #11565-1-AP	IC: 1:400
TRPC1	Intracellular aa’ of human TRPC1	Rabbit Polyclonal	Alomone Labs (Jerusalem, Israel, #ACC-010	IC: 1:500 PLA ^3^: 1:400
TRPC1	C-terminus of human TRPC1	Mouse Monoclonal	Santa Cruz Biotech (Heidelberg, Germany), #sc-133076, Clon E-6	IC: 1:500
TRPC5	Synthetic peptide amino acids 827-845 of human TRPC5	Mouse Monoclonal	Invitrogen (ThermoFisher, Foster City, CA, USA), #MA5-27657, Clon S67-15	IC: 1:500 PLA: 1:400 WB: 1:1000

^1^ Immunocytochemistry; ^2^ Western Blot; ^3^ Proximity ligation assay.

## Data Availability

The datasets generated during the current study are available from the corresponding author upon reasonable request.

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
