# Peer review of "Selective Assembly of TRPC Channels in the Rat Retina during Photoreceptor Degeneration"

_ijms, 2024, doi:10.3390/ijms25137251_

Round 1
Reviewer 1 Report
Comments and Suggestions for Authors
The suggestions are as follows:
Figure 2. F:
What is the unit of Y-axis: per area/square µm (square micrometer) or distance/ µm (micrometer)?
Figure 3.:
In the sentence below, please describe briefly and clearly:
“(B) Bar chart showing that no significant differences in TRPC5 expression appear between the healthy retinas at P23H rat retinas.”
Is it better: “At” -> “and”?
Figure 4. I-K:
Could you show the 3D image of the overlap of muller and astrocytes?
Line 219-222:
“Analysis of whole retinas showed that TRPC1 and STIM1 immunoreactivities were complementary (Figure 5C), with STIM1 distributed around RGC somata, with a typical morphology of Müller cell processes.”
Can the sentence be revised so that it can more clearly state the facts you want to express?
Figure 5:
The colocalization of STIM1/TRPC1 is not obvious. Could you show the 3D images to support the co-localization of the STIM1 and TRPC1?
Line 268-269:
“The various pathological models demonstrated that TRPC1 is a negative regulator of TRPC5 in favor of neuronal protection, and even it has opposite regulatory effects to TRPC5 on neurite outgrowth [42,48].”
Is there a more appropriate or accurate word than “regulator” to use in this sentence? Or modify this sentence to make it more accurate?
Author Response
Estimated reviewer,
I would like to sincerely thank for their constructive comments and suggestions. We have taken into account each of the comments and have made the corresponding modifications to the text, and as a Supplementary Figures (S1, S2 and S3). All the explanations and changes that the manuscript has required are indicated below in in blue. We are convinced that these modifications have significantly strengthened the validity and relevance of the work, and I hope that they will be satisfactory for you.
Sincerely,
Elena Caminos
REVIEWER 1
Open Review
(x) I would not like to sign my review report
( ) I would like to sign my review report
Quality of English Language
(x) I am not qualified to assess the quality of English in this paper
( ) English very difficult to understand/incomprehensible
( ) Extensive editing of English language required
( ) Moderate editing of English language required
( ) Minor editing of English language required
( ) English language fine. No issues detected
|
Yes |
Can be improved |
Must be improved |
Not applicable |
|
|
Does the introduction provide sufficient background and include all relevant references? |
(x) |
( ) |
( ) |
( ) |
|
Is the research design appropriate? |
(x) |
( ) |
( ) |
( ) |
|
Are the methods adequately described? |
(x) |
( ) |
( ) |
( ) |
|
Are the results clearly presented? |
( ) |
(x) |
( ) |
( ) |
|
Are the conclusions supported by the results? |
( ) |
(x) |
( ) |
( ) |
Comments and Suggestions for Authors
The suggestions are as follows:
Figure 2. F:
What is the unit of Y-axis: per area/square µm (square micrometer) or distance/ µm (micrometer)?
Very good question. The unit of the Y-axis is the number of red spots per 210 µm of retina.
We have included this data in the figure legend of FIGURE 2F in the revised manuscript.
Figure 3.:
In the sentence below, please describe briefly and clearly:
“(B) Bar chart showing that no significant differences in TRPC5 expression appear between the healthy retinas at P23H rat retinas.”
Figure 3B is a boxplot chart that provides a wealth of information regarding the distribution of values between animals of the same strain and age (median, first quartile, third quartile, etc.) and the symmetry of the distribution within the same group.
To enhance clarity, we have slightly expanded this description in the figure legend of FIGURE 3B:
“(B) Boxplot chart showing comparison of the relative TRPC5 expression levels between the healthy retinas and P23H rat retinas. No statistically significant differences were observed between the groups, as determined by Kruskal-Wallis’s test and subsequent Dunn’s test”.
However, should the reviewer deem it more meaningful and elucidating to employ another type of diagram, such as a bar diagram representing the mean and standard error of the mean, we would be amenable to modifying the diagram in question.
Is it better: “At” -> “and”? yes, thank you.
Figure 4. I-K:
Could you show the 3D image of the overlap of muller and astrocytes?
3D images do not provide additional information beyond that which ca be discerned in 2D images. This is likely due to the high intensity of immunofluorescence observed in the 3D constructions. Nevertheless, we provide 3D images for the reviewer in supplementary material (Figures S1, S2 and S3), although we believe they are not essential for inclusion in the article.
Line 219-222:
“Analysis of whole retinas showed that TRPC1 and STIM1 immunoreactivities were complementary (Figure 5C), with STIM1 distributed around RGC somata, with a typical morphology of Müller cell processes.”
Can the sentence be revised so that it can more clearly state the facts you want to express?
Thank you. We have revised the aforementioned sentence and included it in the revised manuscript:
“The results of the wholemount retinas analysis indicated that TRPC1 and STIM1 immunoreactivities coexisted, but rarely colocalized, in the cytoplasm of RGCs (Figure 5C). Furthermore, STIM1 was also distributed in glial processes around RGC somata that were immunoreactive to TRPC1.”
Figure 5:
The colocalization of STIM1/TRPC1 is not obvious. Could you show the 3D images to support the co-localization of the STIM1 and TRPC1?
The rationale behind his proposal is sound, given the rarity of colocalization between STIM1 and TRPC1. I regret that you have drawn the conclusion described from the results presented. While both molecules are present in ganglion cells, colocalization has only been observed in exceptional cases. To reinforce these findings, we have included a magnification in a box within Figure 5C.
The figure legend in FIGURE 5 have been changed to explain the incorporation of the magnification images (boxes in Figure 5C):
“Exceptional colocalization of STIM1/TRPC1 in retinal ganglion cells (arrows and magnification in boxes in C, asterisk), and STIM1 immunoreactivity around RGCs (arrowheads).”
Line 268-269:
“The various pathological models demonstrated that TRPC1 is a negative regulator of TRPC5 in favor of neuronal protection, and even it has opposite regulatory effects to TRPC5 on neurite outgrowth [42,48].”
Is there a more appropriate or accurate word than “regulator” to use in this sentence? Or modify this sentence to make it more accurate?
We have revised the sentence:
“Therefore, the various pathological models demonstrated opposite effects of TRPC1 and TRPC5 in neurodegenerative diseases and neurite outgrowth [42,48]. It has been proposed that the overexpression of TRPC1 and inhibition of TRPC5 may help reduce the neurodegeneration effects.”
Despite the limitations of this study, the significance of the experimental model in the field of calcium channels and their relationship with neurodegeneration is evident by the demonstrated interaction of TRPC1 and TRPC5.
I would like to express my sincerest gratitude for your constructive comments, which have significantly enhanced the quality and clarity of the article, in our opinion. Your assistance is greatly appreciated.
Best regards,
Elena Caminos

Reviewer 2 Report
Comments and Suggestions for Authors
The inner retina exhibits significant morphological changes in response to loss of photoreceptor cells, indicating that second-order neurons rely on these cells for their survival. For instance, protecting mitochondria recovers retinal function and promotes the survival of both photoreceptor cells and inner retinal second-order neurons. The present study aims to demonstrate that increased physical interaction between TRPC1 and TRPC1 may contribute to slowing the degeneration of the inner retina during the outer retinal degeneration. In other words, increasing calcium homeostasis plays a role in the inner retina survival after phoreceptor death.
This paper is mainly descriptive without any causative relation between the different events described here
Many information are lacking:
1. Retinal TRP channels have not been properly introduced and recent discussion about their role in retinal health and disease has not been reported
2. The added value of the present study in respect to other studies investigating the role of calcium homeostasis in RP should be better elucidated.
3. The IR study should be accompanied by ERG/PERG studies
4. Whether impeding the interaction among the channels would prevent the delayed degeneration of the inner retina should be ivestigated
5. The role of RGCs should be elucidated mostly in respect to their projection to the visual cortex. In the present study, this point has been emphasized and should be corroborated by VEP recordings
6. The translational potential of the present results should be discussed
7. IR in coronal sections creates a big confusion. Whole mounts even in confocal microscopy would be better together with a schematic representation of each result.
8. WB analysis is rather poor
Comments on the Quality of English Language
English is quite fine
Author Response
Estimated reviewer,
I would like to sincerely thank for their constructive comments and suggestions. We have taken into account each of the comments and have made the corresponding modifications to the text, and as a Supplementary Figures (S1, S2 and S3). All the explanations and changes that the manuscript has required are indicated below in in blue. We are convinced that these modifications have significantly strengthened the validity and relevance of the work, and I hope that they will be satisfactory for you.
Sincerely,
Elena Caminos
REVIEWER 2
Open Review
( ) I would not like to sign my review report
(x) I would like to sign my review report
Quality of English Language
( ) I am not qualified to assess the quality of English in this paper
( ) English very difficult to understand/incomprehensible
( ) Extensive editing of English language required
( ) Moderate editing of English language required
(x) Minor editing of English language required
( ) English language fine. No issues detected
|
Yes |
Can be improved |
Must be improved |
Not applicable |
|
|
Does the introduction provide sufficient background and include all relevant references? |
( ) |
( ) |
(x) |
( ) |
|
Is the research design appropriate? |
( ) |
( ) |
(x) |
( ) |
|
Are the methods adequately described? |
( ) |
( ) |
(x) |
( ) |
|
Are the results clearly presented? |
( ) |
( ) |
(x) |
( ) |
|
Are the conclusions supported by the results? |
( ) |
( ) |
(x) |
( ) |
Comments and Suggestions for Authors
The inner retina exhibits significant morphological changes in response to loss of photoreceptor cells, indicating that second-order neurons rely on these cells for their survival. For instance, protecting mitochondria recovers retinal function and promotes the survival of both photoreceptor cells and inner retinal second-order neurons. The present study aims to demonstrate that increased physical interaction between TRPC1 and TRPC1 may contribute to slowing the degeneration of the inner retina during the outer retinal degeneration. In other words, increasing calcium homeostasis plays a role in the inner retina survival after phoreceptor death.
This paper is mainly descriptive without any causative relation between the different events described here
Many information are lacking:
- Retinal TRP channels have not been properly introduced and recent discussion about their role in retinal health and disease has not been reported
In response to the reviewer's comment, it has been deemed crucial to include pertinent references that examine the function, expression, and physiology of TRP channels in the retina. These references are of particular relevance to those engaged in research on calcium channels. In both the introduction and the discussion of this paper, the role of the TRPC1 and TRPC5 channels in the retina and brain, in health conditions and in several pathological models, has been considered. Given the extensive scope of the topic (seven subfamilies, each with specific functions and distinct activation mechanisms contributing to the functional diversity of TRP channels in the retina), we did not consider discussing other TRP channels in detail. Undoubtedly, this topic is of great relevance due to the significant prospects offered by the increasingly better understanding of the molecular mechanisms, expression, and function of the TRP family in the vertebrate retina.
The following references have been incorporated into the revised manuscript:
- Thébault, 2021
- Križaj et al., 2023
- The added value of the present study in respect to other studies investigating the role of calcium homeostasis in RP should be better elucidated.
We appreciate your input and have incorporated it into the revised manuscript, which now includes a more comprehensive discussion of this topic.
The theoretical basis of the present work is the high concentrations of calcium in neurons and glial cells in the inner retina and in the photoreceptors of P23H animals. These studies are incorporated in the discussion and introduction in the initial version of the manuscript. However, in response to the reviewer's recommendation, we have expanded point 3.1 of the discussion in the revised manuscript (Line 274-290).
- The IR study should be accompanied by ERG/PERG studies
The P23H rat is an experimental model that has been extensively studied and characterized in terms of its visual physiology (ERG, OCT...). The primary work on the electrophysiological characterization of the P23H rat was conducted by Professor LaVail's laboratory at the University of California, with input from our laboratory on evoked potentials and calcium recordings. Several studies have contributed to the ERGs of the P23H-1 strain (Segura et al., 2015; Hanif et al., 2016). However, it is our opinion that including these records in this work would not be justified.
- Whether impeding the interaction among the channels would prevent the delayed degeneration of the inner retina should be investigated
This is a particularly intriguing and pivotal point that we have been contemplating. In fact, we are considering this objective as a potential avenue for future research. However, we are also mindful of the difficulty associated with this proposal. We need to use P23H rats, which would have an additional mutation to the one they have (making animals with a deletion of TRPC1 or TRPC5). Addressing the issue by blocking or inactivating one of these channels is less certain since there are no specific inhibitors. If the reviewer has another way to address this problem, we would be interested in pursuing a collaboration.
- The role of RGCs should be elucidated mostly in respect to their projection to the visual cortex. In the present study, this point has been emphasized and should be corroborated by VEP recordings
This is another interesting point published in several articles. While the potentials evoked by retinal electrical stimulation do not decrease in P23H rats (Koo and Weiland, 2022), the light-evoked activity decreases in RP rats (Wang et al., 2018). These results make sense; in RP, photoreceptors disappear, so light-evoked activity should decrease, but not electrical stimulation-evoked activity, since there are many ganglion cells still connected to the thalamus.
We have included information on this topic in the discussion (at the end of section 3.1; Line 308-311), including a new reference (Koo and Weiland, 2022).
- The translational potential of the present results should be discussed
The discussion of this manuscript has now concluded with a proposal regarding the translational potential of the results. It is of great importance to maintain the inner retina in as intact a state as possible, not only for the purpose of sustaining brain activity, but also for the maintenance of circadian cycles in individuals who have become totally blind. To date, there are no neuroprotective treatments for the inner retina that have taken into account the activity of TRPC calcium channels. In light of our findings, it would be beneficial to ascertain the progression of RP following the inhibition of TRPC1 or/and TRPC5 in RP experimental models, in order to determine the potential therapeutic efficacy of this approach.
- IR in coronal sections creates a big confusion. Whole mounts even in confocal microscopy would be better together with a schematic representation of each result.
Providing results in coronal sections allows for more precise localization of layered immunoreactivity, which is an essential aspect of retinal immunolocalization work. The use of coronal sections is particularly crucial in this process, as they represent descriptions that are being made for the first time. The montage of the whole retina also provides valuable information, but in our study, it serves to supplement and/or complete the information obtained in the coronal sections. In response to the reviewer's request, we have included in the supplementary material a collection of 3D images of both retinal sections and whole retinas, which will help to verify the best identification of immunoreactive elements. Supplemental Figures S1, S2 and S3.
- WB analysis is rather poor
Thanks for this observation. We've added a more detailed explanation of the WB results in the revised manuscript, as well as in the legend for Figure 3. The WB study was designed to confirm what we saw in the TRPC5 immunocytochemistry between healthy and RP animals. Upon examination of the coronal sections of the different retinas, it was observed that TRPC5 exhibited a highly uniform distribution across the various strains and age groups. The WB analysis confirmed that TRPC5 expression remained consistent, thereby corroborating the objective of this study.
Comments on the Quality of English Language: English is quite fine
Despite the limitations of this study, the significance of the experimental model in the field of calcium channels and their relationship with neurodegeneration is evident by the demonstrated interaction of TRPC1 and TRPC5.
I would like to express my sincerest gratitude for your constructive comments and suggestions, which have significantly enhanced the quality and clarity of the article, in our opinion. Your assistance is greatly appreciated.
Best regards,
Elena Caminos

Round 2
Reviewer 1 Report
Comments and Suggestions for Authors
The revision makes some figures and texts related of the manuscript clearer for readers to understand.
Reviewer 2 Report
Comments and Suggestions for Authors
The paper can be accepted
Comments on the Quality of English Language
No comments